# Mechanistic Insight into the Enantioselective Degradation of Esterase QeH to (*R*)/(*S*)–Quizalofop–Ethyl with Molecular Dynamics Simulation Using a Residue-Specific Force Field

**DOI:** 10.3390/ijms25189964

**Published:** 2024-09-15

**Authors:** Yu-Meng Zhu, Gui Yao, Song Shao, Xin-Yu Liu, Jun Xu, Chun Chen, Xing-Wang Zhang, Zhuo-Ran Huang, Cheng-Zhen Xu, Long Zhang, Xiao-Min Wu

**Affiliations:** 1Anhui Province Key Laboratory of Pollutant Sensitive Materials and Environmental Remediation, College of Life Sciences, Huaibei Normal University, Huaibei 235000, China; 12211070750@chnu.edu.cn (Y.-M.Z.); 12007110659@chnu.edu.cn (G.Y.); 20211502049@chnu.edu.cn (S.S.); 12211070744@chnu.edu.cn (X.-Y.L.); 20211502068@chnu.edu.cn (J.X.); zhangxw@chnu.edu.cn (X.-W.Z.); hzrds@hotmail.com (Z.-R.H.); xucz@chnu.edu.cn (C.-Z.X.); 2Institute of Biomedicine, Jinan University, Guangzhou 510632, China; chenchun_jnu@sohu.com; 3School of Computer Science and Technology, Huaibei Normal University, Huaibei 235000, China

**Keywords:** catalytic kinetics characterization, quizalofop–ethyl, enantioselective degradation mechanism, inter-residue interaction, molecular dynamics simulation

## Abstract

The enantioselective mechanism of the esterase QeH against the two enantiomers of quizalofop–ethyl (QE) has been primitively studied using computational and experimental approaches. However, it is still unclear how the esterase QeH adjusts its conformation to adapt to substrate binding and promote enzym*e–substrate interactions in the catalytic kinetics. The equilibrium processes of enzyme–subst*rate interactions and catalytic dynamics were reproduced by performing independent molecular dynamics (MD) runs on the QeH-(*R*)/(*S*)-QE complexes with a newly developed residue-specific force field (RSFF2C). Our results indicated that the benzene ring of the (*R*)-QE structure can simultaneously form anion–π and cation–π interactions with the side-chain group of Glu328 and Arg384 in the binding cavity of the QeH-(*R*)-QE complex, resulting in (*R*)-QE being closer to its catalytic triplet system (Ser78-Lys81-Tyr189) with the distances measured for the hydroxyl oxygen atom of the catalytic Ser78 of QeH and the carbonyl carbon atom of (*R*)-QE of 7.39 Å, compared to the 8.87 Å for (S)-QE, whereas the (*S*)-QE structure can only form an anion–π interaction with the side chain of Glu328 in the QeH-(*S*)-QE complex, being less close to its catalytic site. The computational alanine scanning mutation (CAS) calculations further demonstrated that the π–π stacking interaction between the indole ring of Trp351 and the benzene ring of (*R*)/(*S*)-QE contributed a lot to the binding stability of the enzyme–substrate (QeH-(*R*)/(*S*)-QE). These results facilitate the understanding of their catalytic processes and provide new theoretical guidance for the directional design of other key enzymes for the initial degradation of aryloxyphenoxypropionate (AOPP) herbicides with higher catalytic efficiencies.

## 1. Introduction

Aryloxyphenoxypropionate (AOPP) herbicides are highly active, low toxic, and play an irreplaceable role in increasing production and ensuring agriculture harvests. There are more than 20 kinds of commercially available AOPP herbicides, which are widely used to control annual and perennial grasses in broad-leaved crop fields [1]. In 2014, the global sales of AOPP herbicides accounted for 1.9% of the global pesticide market. However, due to the large-scale application, the environmental pollution problems caused by AOPP herbicides are also becoming increasingly serious. Several studies have shown that the residues of AOPP herbicides can pose potential harm to aquatic organisms or even humans, in addition to causing phytotoxicity to subsequent crops, for example, resulting in liver damage [2], reproductive toxicity [3] and genetic toxicity [4]. Therefore, the dissipation of the AOPP herbicides in the environment is of widespread concern.

Quizalofop–ethyl (QE) is one of the most widely used chiral AOPP herbicides. Commercially available QE takes the form of racemic mixtures (*Rac*-QE), containing equimolar amounts of (*R*)-QE and (*S*)-QE. Of the two enantiomers of QE, the herbicidal activity mainly comes from the (*R*)-enantiomer [5], and (*R*)-QE is mainly used for the purpose of weeding, which is achieved by acting on acetyl-CoA carboxylase to inhibit fatty acid synthesis [6]. Due to its wide range of use, QE residues represent a serious threat to the safety of subsequent crops and other non-target organisms. In addition, for chiral compounds, different enantiomers may have different biological activity and ecological toxicity, and organisms may have three-dimensional selectivity for the absorption and biotransformation of chiral compounds [7,8]. Moreover, recent studies have shown that (*R*)- and (*S*)-enantiomers can even be interconverted under certain conditions [9,10]. Therefore, it is of great significance to study the degradation and transformation of different isomers in the environment.

Biodegradation is the main form of xenobiotic dissipation in the environment, of which microorganisms are the main force. Most of the reported microorganisms that degrade AOPP herbicides are bacteria, which are mainly metabolized by esterase hydrolysis. Moreover, several AOPP herbicide hydrolases have also been identified in various microorganisms. For example, AfeH was identified from the *Acinetobacter* sp. DL-2 strain [11]; ChbH was identified from the *Pseudomonas azotoformans* QDZ-1 strain [12]; Feh was identified from the *Rhodococcus ruber* JPL-2 [13] and *Rhodococcus* sp. T1 strains [14]; and QpeH was identified from the *Pseudomonas* sp. J-2 strain [15]. Recently, an enantioselective esterase, QeH, that preferentially hydrolyzes (*R*)-QE rather than (*S*)-QE was identified from the *Sphingobium* sp. QE-1 strain, as shown in Figure 1 [16]. However, most of the studies on AOPP herbicide hydrolases are concerned with exploring their enzymatic characteristics and substrate spectra, while there are few studies on the enantioselective degradation mechanisms of the different enantiomers of AOPP herbicides. Although Zhou et al. [16] reported that QeH is an enantioselective esterase, they clarified the enantioselective mechanism only according to the docking results based on the three-dimensional structure models of the enzyme and the substrate. It still remains unclear how QeH adjusts its conformation to adapt to substrate binding and facilitates enzyme–substrate interactions during the catalytic dynamic process. Molecular docking can only obtain the static interaction models of enzyme-catalyzed complexes. Since this approach does not take the structural dynamics into account, it has no guarantee of a global minimum. For the PsaA system, its swep-bounded complex was extremely unstable in its docking position during our recent MD study, and it underwent two-stage fluctuations toward its unbounded state due to the unfavorable interactions between both states. How QeH adjusts its conformation and adapts to substrate binding as well as facilitating protein–ligand interactions must still be further elucidated by performing MD simulations [17].

The MD reliability depended on the accuracy with which the interactions between atoms are calculated [17]. Based on the Protein Data Bank (PDB), the pre-residue local conformational preference (coil library) was first constructed by Wu et al. [18]. Through statistical analysis of the coil library, it was found that there were significant differences between the intrinsic conformational preferences of residues and the conformational distributions (backbone dihedral angles *φ*, *ψ* and side-chain torsion angles *χ_i_*) given by conventional force fields. The optimization and correction of pre-residue conformational preferences were achieved by optimizing the dihedral parameters and introducing special local non-bonded interactions [18,19]. Therefore, Wu et al. [18,19,20] designed and developed a range of residue-specific force fields (RSFFs) that can describe binding dynamics, thermodynamics, and intermolecular forces more accurately at the atomic level. In this study, a newly developed residue-specific force field (RSFF2C) with high accuracy and efficiency was adopted to simulate the enantioselective catalytic dynamics process and thermodynamic properties of QeH against the different enantiomers of QE. Through our theoretical calculations, including molecular dynamics simulations (MD), MM/PBSA, and computational alanine scanning (CAS) methods, the enantioselective degradation mechanism of QeH against (*R*)/(*S*)-QE was further elucidated. This study will enrich the theoretical research on the enantioselective degradation of chiral herbicides by microorganisms and also provide new theoretical guidance for the directional design of a high-catalytic-efficiency AOPP herbicide-initiated degradation enzyme.

## 2. Results and Discussion

### 2.1. Stability and Dynamics Analysis of the MD Simulation

The QeH structure models predicted by Alphafold2 were evaluated using the PROCHECK [21,22] and Verify 3D programs, with the results shown in Figure 2. The Ramachandran plot illustrates the allowed and disallowed conformations of amino acid residues in proteins, where the red regions indicated the most favored areas, the yellow regions represented generously allowed areas, and the blank regions were disallowed areas. As shown in Figure 2A, all the residues of the QeH model were absent from the disallowed regions, consistent with the stereochemical energy rules. The Verify 3D program assessed the compatibility of the protein’s 3D model with its amino acid sequence. As shown in Figure 2B, 99.54% of the amino acid residues scored above 0.1, meeting the evaluation criteria. Combining these analyses with the high confidence (predicted local distance difference test, pLDDT) [23,24] scores from AlphaFold2 (Table 1), the predicted Rank_0 structure was confirmed to be reasonable and suitable for further calculations.

The docking results for ten independent QeH and (*R*)/(*S*)-QE complexes are summarized in Table 2. Conformational cluster analysis of these docked complexes showed that their binding poses belonged to the same type of conformation and ten representative snapshots of (*R*)/(*S*)-QE superposed inside the interior of their respective hydrophobic cavity. It was found that the binding energies of esterase QeH to (*R*)-QE and (*S*)-QE corresponding to the preferred pose were calculated to be −9.26 and −8.79 kcal·mol^−1^, respectively. It is obvious that the binding affinity of QeH to (*R*)-QE was slightly higher than that of (*S*)-QE. For determining the details about their binding stability, catalytic dynamics, and the enzymatic catalytic interaction mode under the equilibrated state, independent 500 ns MD simulations were performed for the QeH-(*R*)-QE and QeH-(*S*)-QE systems using a residue-specific force field (RSFF2C).

As shown in Figure 3A,B, the RMSD curve of the QeH-(*R*)-QE system fluctuated within the initial 150 ns, and then it was equilibrated at ca. 1.8 Å with a standard deviation (SD) of 0.13 Å, whereas the time evolution RMSD values of the QeH-(*S*)-QE system underwent several large fluctuations and reached equilibrium at 300 ns, with its average and SD value being ca. 3.5 and 0.13 Å, respectively. It was obvious that QeH and (*R*)-QE/(*S*)-QE were stable after 150 and 300 ns of MD simulations, respectively, and these equilibrated MD stages were used for further trajectory analysis. However, the fluctuation of the QeH-(*S*)-QE complexed system was significantly larger than that of the QeH-(*R*)-QE system. For the QeH-(*R*)-QE complex, the RMSD curve of QeH rapidly increased to ca. 2.0 Å at the initial stage of the MD simulation and then synchronously reached equilibrium around ca. 1.7 Å afterwards. For the QeH-(*S*)-QE system, the RMSD curve of QeH sharply rose to ca. 3.8 Å until 140 ns, and then the curve gradually stabilized at ca. 3.5 Å. In addition, the RMSD curve of (*R*)-QE remained stable at ca. 0.6 Å after the initial 50ns rapid fluctuation, while the RMSD curve of (*S*)-QE fluctuated at ca. 1.0 Å during the process of MD simulation. From the perspective of the ligand, this also indicated that (*R*)-QE was more stable than (*S*)-QE in the QeH-(*R*)/(*S*)-QE systems.

By calculating the root-mean-square fluctuation (RMSF) of these two complexed systems, the flexibility changes of QeH during the MD runs were analyzed [25]. The averaged RMSF and standard deviation for all the residues and after removing two terminal residues are listed in Table 3. As listed in Table 3 and shown in Figure 3C,D, the protein flexibility of QeH in the QeH-(*R*)-QE system was lower than that in the QeH-(*S*)-QE system. Conformational cluster analysis was performed on the whole trajectory of independent MD runs using the RMSD based on the protein backbone as the distance metric. The QeH-(*R*)-QE complex was stable with one predominant conformation cluster, while the QeH-(*S*)-QE complex was clustered into three dominant conformations and displayed great volatility. To further analyze the dynamic characteristics of these two complexes, the ten equilibrium conformations of these two systems during the independent MD simulation were superimposed to their initial docking structures, respectively. The typical conformations of (*R*)-QE and (*S*)-QE were superimposed at the active site of the QeH binding cavity and are shown in Figure 4A,B. (*R*)-QE and (*S*)-QE were located in the QeH binding cavity and did not undergo a large degree of conformational transition throughout the process from the initial docking to the MD simulation. However, it can be seen in Figure 4 that the side chain of (*S*)-QE showed a certain degree of instability during the MD simulation, and it was not as tightly bound to the QeH active cavity as (*R*)-QE. These calculation results were in excellent agreement with our previous experimental results on QeH catalyzing chiral QE [16].

### 2.2. Thermodynamics Analysis of the MD Simulation

Calculating the binding free energy of biological macromolecules is an effective method to characterize intermolecular binding affinity, which has been widely used in the field of biomolecular simulation [17,25,26,27,28,29]. The total binding free energy (∆*G_bind_*) values of the QeH-(*R*)-QE and QeH-(*S*)-QE complexes were calculated from the equilibrium trajectory of the independent MD run and are listed in Table 4. The total binding energy of the QeH-(*R*)-QE complexed system from the equilibrated conformations was calculated as −40.94 ± 3.7851 kcal·mol^−1^. However, for the QeH-(*S*)-QE complex, the ∆*G_bind_* value of the equilibrium trajectory was calculated to be −38.95 ± 3.7882 kcal·mol^−1^. After the *t*-test, the difference in the ∆*G_bind_* value between these two QeH systems was statistically significant (*n_R_* = 351, *n_S_* = 201, *t* = 2.44, *p* = 0.015). The four energy decomposition item values involving van der Waals (∆*G_vdw_*), electrostatic (∆*G_ele_*), polar, and apolar solvation energies (∆*G_PB_* and ∆*G_Surf_*) represented different contributions to the binding affinity of the QeH-(*R*)-QE and QeH-(*S*)-QE complexes. The ∆*G_vdw_* values for the QeH-(*R*/*S*)-QE systems were calculated to be −51.46 ± 3.25 and −51.77 ± 2.54 kcal·mol^−1^, respectively, which were much higher than other energy decomposition items. The values of apolar solvation items were quite low, with the ∆*G_Surf_* values being −5.81 ± 0.26 and −6.07 ± 0.22 kcal·mol^−1^, respectively. The ∆*G_PB_* values for these two QeH-(*R*/*S*)-QE systems were 36.26 ± 2.92 and 34.07 ± 4.08 kcal·mol^−1^, respectively. It was worth noting that the ∆*G_ele_* values for QeH-(*R*/*S*)-QE complexes were −19.92 ± 4.10 and −15.18 ± 4.81 kcal·mol^−1^, respectively. It indicated that the van der Waals, electrostatic, and aploar interactions played important roles in the QeH-(*R*/*S*)-QE catalysis processes. The strong polar solvation energy decomposition item played an unfavorable role in the binding processes of the QeH-(*R*)-QE and QeH-(*S*)-QE complexes. Due to the stronger electrostatic interactions and stabilities, QeH could bind to the substrate (*R*)-QE and much closer to its catalyzing triplet system (Ser78-Lys81-Tyr189). By performing per-residue free energy decomposition (PFED) calculations [25,26,27,28], the important residues involved in the binding cavity of the QeH-(*R*/*S*)-QE complexed systems were obtained. The contributions of ∆*G_bind_* and four energy decomposition items (∆*G_vdw_*, ∆*G_ele_*, ∆*G_PB_* and ∆*G_Surf_*) of each residue to the binding interactions between QeH and (*R*/*S*)-QE are displayed in Figure 5 and Table 5. The key residues (Tyr331, Tyr350, Trp351, Gly352 and Arg384) with their respective binding affinities over −1.0 kcal·mol^−1^ were identified between QeH and (*R*)-QE, and their ∆*G_bind_* values were calculated as −1.84, −2.71, −3.59, −1.01 and −3.78 kcal·mol^−1^, respectively. In particular, Tyr331, Tyr350, Trp351, and Arg384 contributed significantly to its binding stability, whereas the key residues for the binding affinity of the QeH-(*S*)-QE system involved Tyr189, Phe326, Glu328, Tyr350, Trp351 and Val354, and their binding free energies were −1.13, −1.26, −1.75, −1.27, −3.91 and −1.94 kcal·mol^−1^, respectively (Figure 4). It can be seen in Figure 5B that due to the swinging of the (*S*)-QE side chain, the key residues exhibit a weaker binding affinity between QeH and (*S*)-QE. In addition, the ∆*G_bind_* contributions of the two ligands in the QeH-(*R*/*S*)-QE complexed systems were also obtained, which were calculated to be −21.86 ± 1.96 and −20.31 ± 1.88 kcal·mol^−1^, respectively. The differences in the energy decomposition items were similar to the total binding energies of these two complexes. Further, the electrostatic interaction of (*R*)-QE was stronger than that of (*S*)-QE, and their electrostatic energies were calculated to be −9.96 ± 2.05 and −7.59 ± 2.40 kcal·mol^−1^, respectively.

### 2.3. The Enantioselective Degradation Mechanism of (R/S)-QE by QeH

Since the carbon atom of the propionic acid group connected to the phenoxy group was an asymmetric carbon atom, most AOPP herbicides had optical activity; that is, there were two chiral isomers. It was shown that the activity of the (*R*)-enantiomer of AOPP herbicides was almost twice that of their enantiomers [30]. Zhou et al. described the experimental phenomenon of the selective degradation of (*R*/*S*)-QE catalyzed by QeH isolated from the *Sphingobium* sp. QE-1 strain [16]. Our MD results allowed us to decipher the enantioselective degradation mechanism of (*R*/*S*)-QE catalyzed by QeH and identify the key residues in their respective catalytic processes.

It can be seen in Figure 4 and Figure 6A,C that (*R*)-QE was steadily anchored inside the binding cavity formed by the nonpolar residues Phe326, Ala329, Val334, Trp351, Val354, and Phe358 and polar residues such as Glu328, Tyr331, Tyr350, Gly352, and Arg384. In addition to these important hydrophobic and hydrogen bonding interactions, the π-π stacking interaction formed between the side chain groups of Tyr331 and Trp351 and the benzene ring of (*R*)-QE affected and contributed significantly to the binding ability of the QeH-(*R*)-QE complex. Although (*S*)-QE was also bounded in the same active cavity, its benzene ring only formed a π-π stacking interaction with the side chain of Trp351. In addition, the differences in electrostatic interactions between the QeH and (*R*/*S*)-QE systems are also shown in Figure 6B,D. During the MD simulation process, the cation-π and anion-π interactions simultaneously formed between the benzene ring of (*R*)-QE and the side chains of Glu328 and Arg384 meant that (*R*)-QE was closer to the catalytic triplet (Ser78-Lys81-Tyr189) with the distances measured for the hydroxyl oxygen atom of the catalytic Ser78 of QeH and the carbonyl carbon atom of (*R*)-QE of 7.39 Å, compared to the 8.87 Å for (*S*)-QE Figure 7 and Table 6 [16]. However, Glu328 only had anion-π interaction with (*S*)-QE, so that (*S*)-QE was less close to its catalytic site. It suggested that (*R*)-QE could stably bind to QeH and was preferentially degraded by forming stronger π-π stacking and ion-π interactions with the residues. The enantioselective degradation of QeH is due to its relatively weak interaction with (*S*)-QE (Table 6).

As shown in Figure 8, the ester bond hydrolysis process of (*R*)-QE catalyzed by esterase QeH can be decomposed into three steps. The anion-π, cation-π, and π-π stacking interactions involved by the identified key amino acids of QeH facilitated (*R*)-QE to bind to the catalytic Ser78, and its hydroxyl hydrogen atom was transferred to Lys81 to form an oxygen anion and attacked the carbonyl carbon atom of (*R*)-QE to form a C-O bond. Then, the hydrogen atom of Tyr189 was transferred to the ester oxygen atom of (*R*)-QE, and a hydrogen atom from Lys81 simultaneously removed an ethanol molecule for Tyr189. Finally, the ester bond formed by (*R*)-QE and Ser78 was hydrolyzed under the action of water molecules, and the catalytic active site was reduced to the initial state. (*R*)-QE was hydrolyzed into two parts: carboxyl group and hydroxyl group.

To further investigate the possibility of these identified residues participating in the catalyzed activities of QeH-(*R*)-QE and QeH-(*S*)-QE, respectively, we performed CAS calculations for these two complex systems, and the CAS results are listed in Table 7. The alanine substitution at these identified residues (Glu328, Tyr350, Trp351, Gly352 and Arg384) of the QeH-(*R*)-QE complex resulted in their mutation energy (∆∆*G_mut_*) values exceeding the threshold (0.5 kcal·mol^−1^) [25,26,27,28,31]. The residues of the QeH-(*S*)-QE complex with mutation energies exceeding 0.5 kcal·mol^−1^ included Tyr189, Phe326, Glu328, Tyr350, and Trp351. In particular, the ∆∆*G_mut_* values of the Trp351 residue could reach as high as 2.01 and 2.02 kcal·mol^−1^ in the QeH-(*R*/*S*)-QE systems, respectively. Trp351 was ascertained to be a hotspot residue for participating in the QeH-(*R*)-QE and QeH-(*S*)-QE catalyzed activities due to the formed π-π stacking interaction between them. Alanine substitutions at the hotspot site could severely weaken this interaction and greatly affect the catalytic activity of QeH. In addition, the hotspot residue Glu328 was closely related to the formation of the ion-π interaction, and the mutation energies showed obvious differences in the QeH and (*R*/*S*)-QE complexes, which were 1.27 and 0.89 kcal·mol^−1^, respectively. It indicated that the mutation of the Glu328 in the QeH-(*R*)-QE complex imposed a greater effect on its catalytic activity than that of the QeH-(*S*)-QE complex.

## 3. Materials and Methods

### 3.1. Structure Prediction

Since the crystal structure of esterase QeH is not currently available in the Protein Data Bank (PDB), AlphaFold2 (AlphaFold v2.0, DeepMind Technologies Limited, London, United Kingdom) was used to predict the three-dimensional structure of QeH [23]. The structural prediction in this study was conducted on a Linux operating system using a Python 3.9.7 and CUDA 10.2.89 environment, with the configuration of AlphaFold2 simplified through Docker scripts. Several databases required by AlphaFold2, including BFD, MGnify, PDB70, PDB, PDB seqres, UniRef30, UniProt, and UniRef90, were pre-downloaded and stored in a designated directory. The target protein esterase QeH’s sequence in FASTA format was prepared and named “QeH.fasta”. AlphaFold2 was then executed for structural prediction using the following command: source activate; conda activate alphafold2; python /opt/alphafold2/docker/run_docker.py --fasta_paths=QeH.fasta --max_template_date=2022-07-06 --db_preset=‘full_dbs’ --data_dir=/home/hipeson/software/MSA --gpu_devices 0. In this command, fasta_paths specifies the path to the FASTA file containing the target protein sequence (in this case, QeH.fasta); max_template_date sets the maximum template date to ensure that the template data used are no later than this date; db_preset selects the model configuration, where ‘full_dbs’ indicates that all genetic databases from CASP14 are used; data_dir specifies the directory where the databases are stored (/home/hipeson/software/MSA); and gpu_devices assigns the GPU device to be used for computation, with 0 indicating the first GPU. The predicted ranker_0 structure, which achieved the highest score of 94.696 (as shown in Table 1), was selected as the structural model of QeH for subsequent calculations [24].

### 3.2. Molecular Docking

Molecular docking was performed using the AutoDock4 (AutoDock v4.2.6, The Scripps Research Institute, Molecular Biology Building, Room 112, 10550 North Torrey Pines Road, La Jolla, CA 92037, USA) software package [32]. The two configurations of (*R*)/(*S*)-QE were separately hydrogenated, and the resulting structures were optimized by the MOPAC program [33]; the PM3 atomic charge was calculated [34]. Finally, AutodockTools (AutodockTools v1.5.6, The Scripps Research Institute, Molecular Biology Building, Room 112, 10550 North Torrey Pines Road, La Jolla, CA 92037, USA) [35] was then used to manipulate the structures of the receptor and ligand separately, so that the docking box could encapsulate the active site. The grid dimensions were set to 60 × 40 × 40 points in the XYZ directions, with a grid spacing of 0.375 Å. The coordinates of the docking box center were (1.03, 1.29, 0.08) based on the active site identified by catalytic triads (Ser78-Lys81-Tyr189). The number of docking runs was set to 10, while the other parameters were kept at default values. The optimal poses were selected as the initial structures for independent molecular dynamics simulations.

### 3.3. Molecular Dynamics (MD) Simulation

The AMBER20 (AMBER v20.0, University of California, San Francisco, CA, USA) software [36] was used to perform independent MD simulations for the QeH-(*R*)-QE and QeH-(*S*)-QE complexed systems with a residue-specific force field (RSFF2C). The QeH structure was verified using DS 2019 (Discovery Studio Client v19.1.0.18287, Dassault Systèmes, Vélizy-Villacoublay, France) software, and all hydrogen atoms were removed [37]. The substrates of (*R*)/(*S*)-QE were preprocessed using the GAFF force field. Electrostatic potential (ESP) charges were calculated at the HF/6-31G* level using Gaussian16 (Gaussian 16 Rev. C.01, Gaussian, Inc., Wallingford, CT, United States) [38], and these charges were fitted to the atomic charges using the restrained electrostatic potential (RESP) procedure in Antechamber. The coordinates of (*R*)*/*(*S*)-QE were then integrated into the QeH structure to form the complex. The force field setup began by generating the initial topology and coordinate files in AmberTools using the ‘tleap’ module based on the AMBER ff14SB force field. The generated topology file was modified by adding CMAP correction terms and other dihedral-angle corrections to implement the RSFF2C force field [18,19,20]. The modified topology file was used for subsequent molecular dynamics simulations. Through these steps, the topology and parameter files for the protein–ligand complex compatible with the RSFF2C + GAFF force fields were successfully generated, providing a reliable foundation for molecular dynamics simulations. Each system was solvated in a TIP3P box filled with ca. 12828 water molecules [39]. The Particle Mesh Ewald (PME) algorithm was applied to calculate the long-range electrostatic interaction [40], and the LINCS algorithm was utilized to constrain the hydrogen atoms [41]. Both the cut-off distances of van der Waals and electrostatic interactions were set to 9.0 Å. The periodic boundary condition (PBC) was adopted to eliminate boundary effects. Two sodium ions were added to each system in order to neutralize the net charge, ensuring a neutral environment. Subsequently, the systems were optimized sequentially by the steepest descent (SD) and conjugate gradient (CG) algorithm prior to our MD simulations. NVT and NPT ensembles were used for position-restricted MD simulations. Through applying Parrinello–Rahman barostats and V-rescale thermostats [42,43], the simulation pressure and temperature were coupled at 1 bar and 300 K. Each independent MD simulation was conducted for 500 ns, and the conformation was saved every 200 ps for the MD trajectory.

### 3.4. MD Trajectory Analysis

The MD trajectories were analyzed using the cpptraj program from AMBER20 software. The binding stability and dynamics of the QeH-(*R*)-QE and QeH-(*S*)-QE complexed systems were assessed by calculating the time-dependent root-mean-squared deviation (RMSD), root-mean-squared fluctuations (RMSF) of the protein system, and key non-bonding interactions as well as principal component analysis (PCA) for the ligands (*R*)-QE and (*S*)-QE. All the structural images were generated using the open-source software PyMOL 2.6.0 (PyMOL Molecular Graphics System, Version 2.6 Schrödinger, LLC*,* New York, NY, USA) and the commercial life science software package DS 2019 [37,44].

### 3.5. Binding Free Energy Calculation

By using the MMPBSA.py.MPI script that comes with the AMBER20 software and adopting the MM/PBSA method, the total and per-residue binding free energy (PFED) of these two herbicides catalyzed by the QeH structure were calculated in parallel via selecting their respective equilibrium stages. It can be expressed by the following equation:∆*G_bind_* = ∆*G_vdw_* + ∆*G_ele_* + ∆*G_PB_* + ∆*G_Surf_* − *T*∆*S*
where ∆*G_bind_* refers to the total binding free energy of the esterase–substrate systems and was calculated by the following terms: ∆*G_vdw_* and ∆*G_ele_* denote the van der Waals and Coulomb electrostatic interaction energies, respectively. ∆*G_PB_* and ∆*G_Surf_* are the solvation free energy differences of polar and apolar, respectively. Given that our study focused on the relative binding free energy between the two enantiomers, the entropic contribution (−*T*∆*S*) was expected to largely cancel out between the two, so it was approximately neglected [45].

### 3.6. Computational Alanine Scanning (CAS)

CAS served as an effective tool for determining the interaction residues of protein–ligand complexes, which can be used to validate the previous results of pre-residue binding free energy decomposition calculation. The mutation energy (∆∆*G_mut_*) was calculated using the Calculate Mutation Energy (Binding) module of DS 2019 according to the following formula:∆∆*G_mut_* = ∆*G_bind_* (*mutant*) − ∆*G_bind_* (*wild*-*type*)
where ∆∆*G_mut_* denoted the binding free energy change of the esterase–substrate systems before and after the amino acid (Ala) substitution; ∆*G_bind_* (*mutant*) and ∆*G_bind_* (*wild-type*) indicated the binding free energy in the mutant and wild-type systems, respectively. If the ∆∆*G_mut_* value was between −0.5 and 0.5 kcal·mol^−1^, the mutation had no effect on the binding affinity; if the value of ∆∆*G_mut_* was greater than 0.5 kcal·mol^−1^, the mutation resulted in a decreased binding affinity; if the ∆∆*G_mut_* value was less than −0.5 kcal·mol^−1^, the mutation resulted in an increased binding affinity [17,25,26,27,28,29].

## 4. Conclusions

In this study, with the aid of the advanced residue-specific force field RSFF2C, we conducted an all-atom MD investigation to comprehensively elucidate the enantioselective degradation dynamics and thermodynamic characteristics of quizalofop-ethyl (QE) enantiomers by esterase QeH. Our findings showed that van der Waals, Coulomb electrostatic, and nonpolar solvation interactions played roles in the chiral catalysis process. QeH preferentially bound and catalytically degraded (*R*)-QE via its stronger electrostatic interaction. Pre-residue binding free energy (PFED) calculations revealed that in the (*R*)-QE complex, the π-π stacking interactions between the phenyl ring of Tyr331 and the indole ring of Trp351 and the phenyl ring of (*R*)-QE significantly contributed to its binding and catalytic degradation, whereas in the QeH-(*R*)-QE complex, the phenyl ring of (*R*)-QE simultaneously formed anionic-π and cation-π interactions with the side chain groups of Glu328 and Arg384 on the esterase, respectively, but in the QeH-(*S*)-QE complex, the phenyl ring of (*S*)-QE formed a π-π stacking interaction with the indole ring of Trp351, and only Glu328 had an anionic-π interaction with (*S*)-QE. It indicated that (*R*)-QE could stably bind to QeH and be preferentially degraded by forming stronger ion-π and π-π stacking interactions with its residues, and the enantioselective degradation behavior of QeH was due to the relatively weak interaction formed with (*S*)-QE.

The computational alanine scanning (CAS) results indicated that the mutation energies of the QeH-(*R*)-QE/QeH-(*S*)-QE complexes, especially that of residue Trp351, reached 2.01 and 2.02 kcal·mol^−1^, respectively. It revealed that the hotspot residue of Trp351 formed π-π stacking interactions in these two complexes to participate in the catalytic process, and after being replaced by alanine, this interaction would be severely weakened and greatly affect the QeH activity in catalyzing (*R*)-QE/(*S*)-QE, which further verified the key role of π-π stacking interactions during the binding and catalytic process of QeH to different chiral isomers of QE. The catalytic activity of Trp351 should be maintained to ensure the stable binding affinity of QeH to different conformations of QE. These calculation results clarified the enantioselective degradation mechanism of quizalofop chiral isomers by esterase QeH, providing an important scientific basis for understanding the causes of the selective degradation of chiral herbicides by microorganisms, offering new theoretical guidance for the directional design of key enzymes that initiate the degradation of AOPP herbicides with higher catalytic efficiency. Our findings provide new theoretical guidance for the directional design of other key enzymes for the initial degradation of aryloxyphenoxypropionate (AOPP) herbicides with higher catalytic efficiencies.

## Figures and Tables

**Figure 1 ijms-25-09964-f001:**
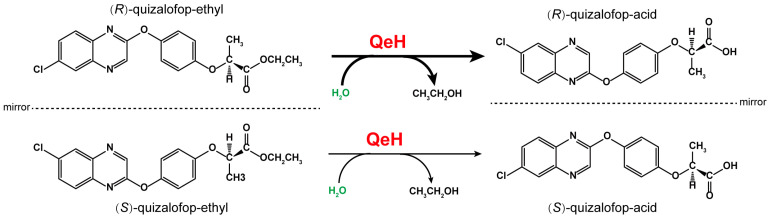
The proposed catabolic pathway of (*R*)–quizalofop–ethyl and (*S*)–quizalofop–ethyl by the esterase QeH.

**Figure 2 ijms-25-09964-f002:**
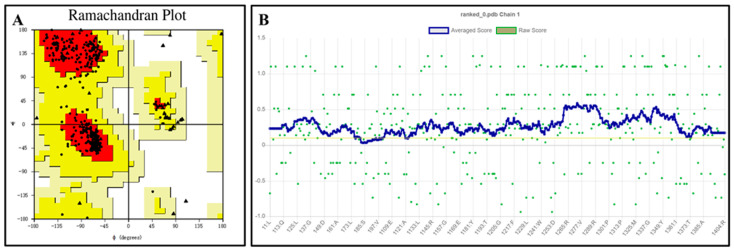
Ramachandran plot (**A**) and verify 3D score (**B**) for the QeH model. The red regions indicated the most favored areas, the yellow regions represented the generously allowed areas, and the blank regions was the disallowed areas. The yellow line in the VERIFY plot (**B**) represented a threshold for the averaged 3D-1D score, specifically at Y = 0.1. This line was used to indicate areas or scores that meet or exceed this threshold value.

**Figure 3 ijms-25-09964-f003:**
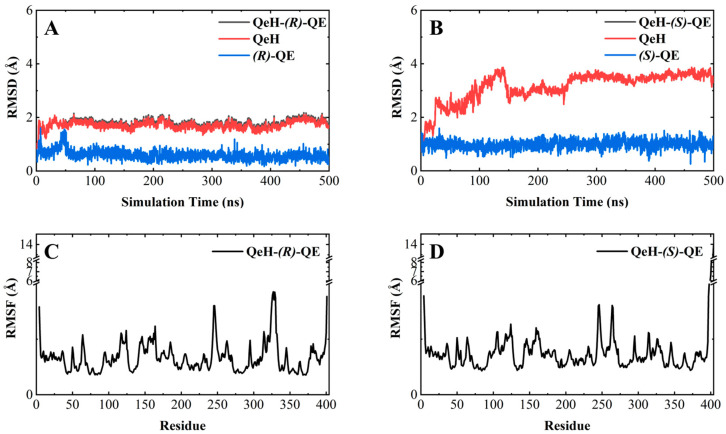
Molecular dynamics simulations of the QeH-(*R*)-QE and QeH-(*S*)-QE complexed systems. The root-mean-square deviation (RMSD) and root-mean-square fluctuation (RMSF) curves of the QeH-(*R*)-QE (**A**,**C**) and QeH-(*S*)-QE (**B**,**D**) complexed systems as functions of simulation time during the MD runs.

**Figure 4 ijms-25-09964-f004:**
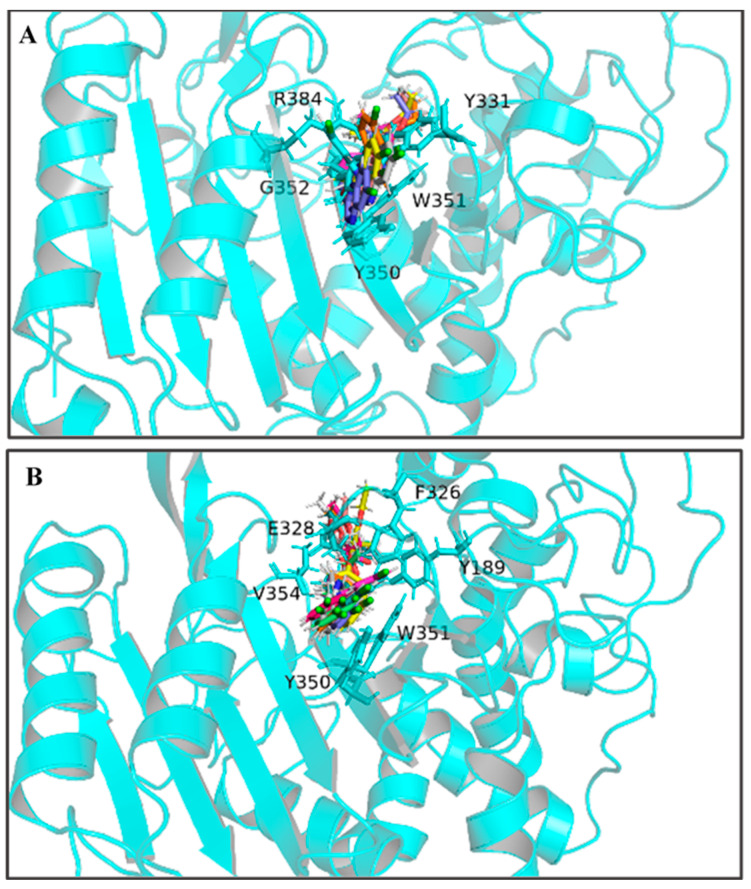
The ten representative snapshots of (*R*)-QE (**A**) and (*S*)-QE (**B**) superposed at their respective QeH active sites inside the interior of hydrophobic pocket during their MD runs. Key residues of QeH and two ligands were represented by stick models, and the residues (Tyr331, Tyr350, Trp351, Gly352 and Arg384 for the QeH-(*R*)-QE complex; Tyr189, Phe326, Glu328, Tyr350, Trp351 and Val354 for the QeH-(*S*)-QE complex) with their respective binding affinities over −1.0 kcal·mol^−1^ were marked by black labels.

**Figure 5 ijms-25-09964-f005:**
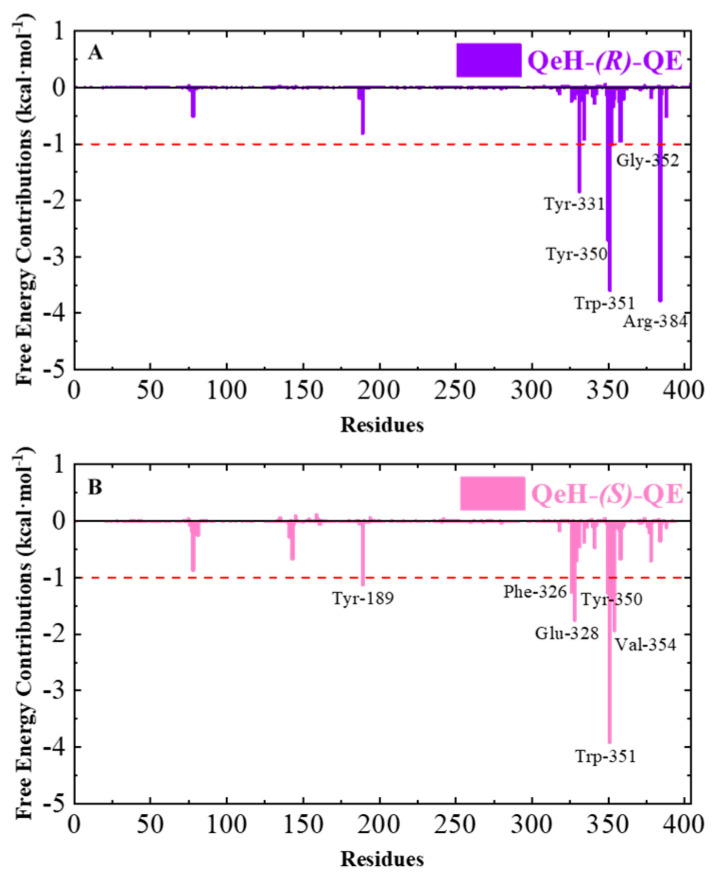
The total binding free energy (∆*G_bind_*) contributions of the QeH-(*R*)-QE (**A**) and QeH-(*S*)-QE (**B**) complexes. Each residue for the QeH-(*R*)-QE and QeH-(*S*)-QE complexes calculated from the equilibrated conformations during independent MD runs. The residues contribution exceeding −1.00 kcal·mol^−1^ to the binding free energy were marked with red dashed lines.

**Figure 6 ijms-25-09964-f006:**
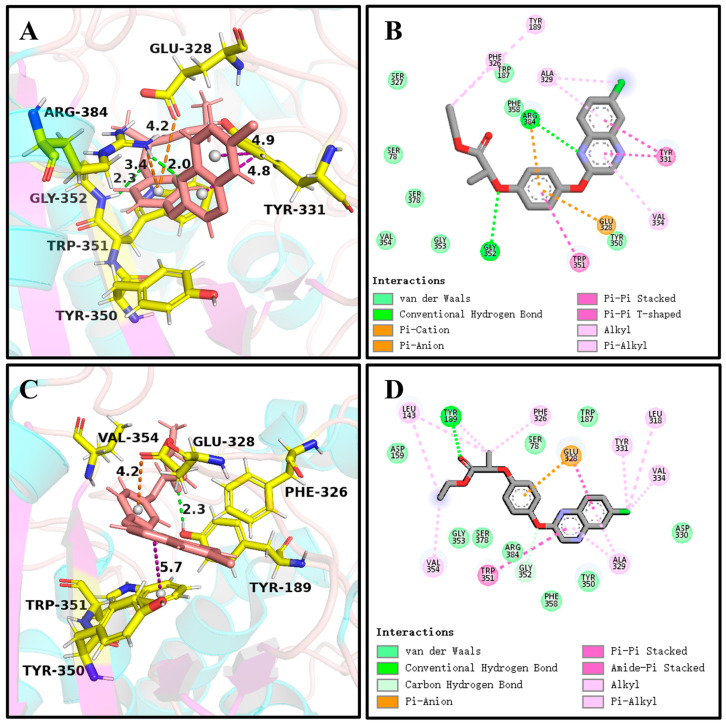
The key interactions at the active sites of the representative conformations of the QeH-(*R*)-QE and QeH-(*S*)-QE complexes with equilibrium stabilization. The interactions derived from the representative conformation of the QeH-(*R*)-QE (**A**,**B**) and QeH-(*S*)-QE (**C**,**D**) complexes generated by the MD simulations were represented by dotted lines in different colors, and the unit of interaction distances was Å.

**Figure 7 ijms-25-09964-f007:**
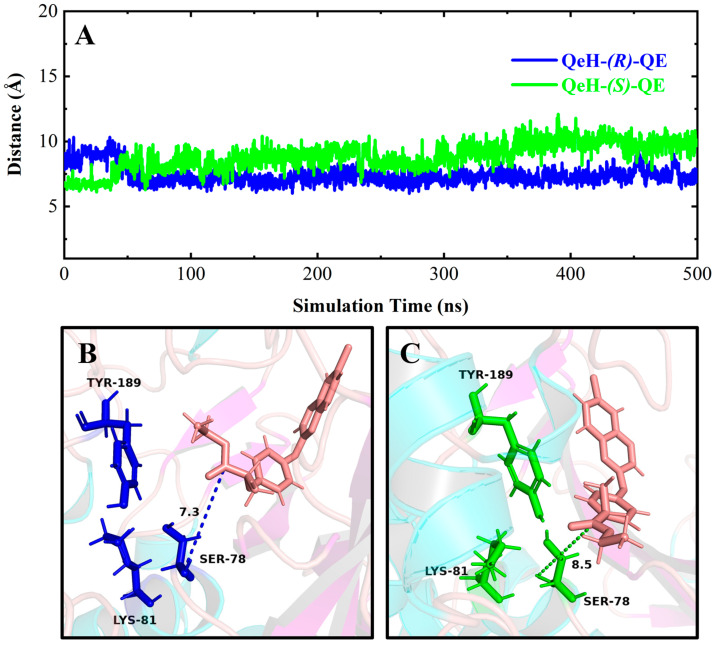
Time-dependent distances between the hydroxyl oxygen atom of the catalytic Ser78 of QeH and the carbonyl carbon atom of (*R*)/(*S*)-QE (**A**), and the representation of the catalytic triad of QeH and the substrate (*R*)/(*S*)-QE (**B**,**C**). The catalytic triad (Ser78, Lys81, and Tyr189) of QeH was in blue and green, whereas the (*R*)/(*S*)-QE was salmon.

**Figure 8 ijms-25-09964-f008:**
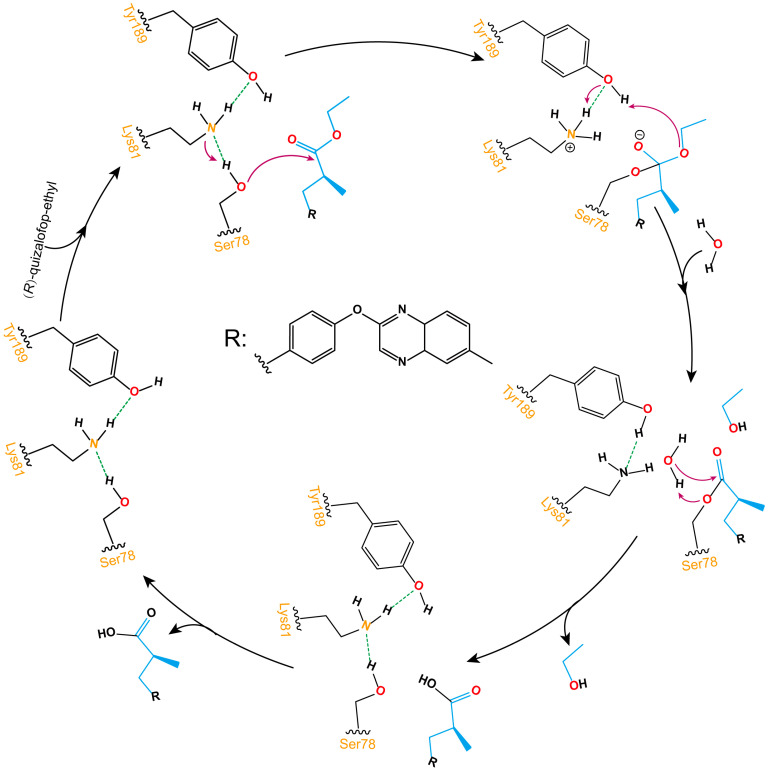
Schematic illustration of the ester bond hydrolysis process of (*R*)-QE catalyzed by esterase QeH. The purple arrows displayed the transfer reaction of hydrogen atoms, and the dotted green lines showed the formation of hydrogen bonds between hydroxyl hydrogen atoms and nitrogen atoms on amino groups.

**Table 1 ijms-25-09964-t001:** The pLDDT scores of structure predictions using AlphaFold2.

Esterase	Structure	Rank Score
QeH	Rank_0	94.696
	Rank_1	94.675
	Rank_2	94.540
	Rank_3	94.322
	Rank_4	94.261

**Table 2 ijms-25-09964-t002:** The docking results of the esterase QeH to (*R*)/(*S*)-QE (measurement unit: kcal·mol^−1^).

Ligand	Structure	Docking Energy	Mean	Standard Deviation
(*R*)-QE	1	−9.26	−7.55	1.87
	2	−9.17		
	3	−8.93		
	4	−8.87		
	5	−8.65		
	6	−8.54		
	7	−6.74		
	8	−6.18		
	9	−4.58		
	10	−4.57		
(*S*)-QE	1	−8.79	−6.98	0.98
	2	−8.29		
	3	−7.09		
	4	−7.04		
	5	−6.91		
	6	−6.89		
	7	−6.78		
	8	−6.67		
	9	−5.73		
	10	−5.60		

**Table 3 ijms-25-09964-t003:** The averaged RMSF and standard deviation for all the residues except for the tail (measurement unit: Å).

System	Residue	Averaged RMSF	Standard Deviation
QeH-(*R*)-QE	all the residues	0.71	0.45
	expect for the tail	0.69	0.34
QeH-(*S*)-QE	all the residues	0.87	1.16
	expect for the tail	0.83	0.93

**Table 4 ijms-25-09964-t004:** The binding free energies (∆*G_bind_*) and each energy item for the QeH-(*R*)-QE and QeH-(*S*)-QE complexes systems calculated from different equilibriums of the RSFF2C MD simulations (measurement unit: kcal·mol^−1^).

System	Time Scope (ns)	Van der Waal Energy (∆*G_vdw_*)	Electrostatic Energy (∆*G_ele_*)	Polar Solvation Energy (∆*G_PB_*)	Apolar Solvation Energy (∆*G_Surf_*)	Binding Energy (∆*G_bind_*)
QeH-(*R*)-QE	150–500	−51.46 ± 3.25	−19.92 ± 4.10	36.26 ± 2.92	−5.81 ± 0.26	−40.94 ± 3.7851
QeH-(*S*)-QE	300–500	−51.77 ± 2.54	−15.18 ± 4.81	34.07 ± 4.08	−6.07 ± 0.22	−38.95 ± 3.7882

**Table 5 ijms-25-09964-t005:** The decomposition for the important residues contributing to the binding free energy of the QeH-(*R*)-QE and QeH-(*S*)-QE complexes systems calculated from equilibrium time scopes of independent MD trajectory (measurement unit: kcal·mol^−1^).

System	Residue	Van der Waal Energy (∆*G_vdw_*)	Electrostatic Energy (∆*G_ele_*)	Polar solvation Energy (∆*G_PB_*)	Apolar solvation Energy (∆*G_Surf_*)	Binding Energy (∆*G_bind_*)
QeH-(*R*)-QE	Tyr331	−2.75 ± 0.54	−0.13 ± 0.39	1.30 ± 0.42	−0.26 ± 0.06	−1.84 ± 0.59
	Tyr350	−2.70 ± 0.58	−0.78 ± 0.42	0.98 ± 0.37	−0.21 ± 0.05	−2.71 ± 0.67
	Trp351	−3.82 ± 0.46	−1.07 ± 0.69	1.53 ± 0.47	−0.23 ± 0.04	−3.59 ± 0.65
	Gly352	−1.18 ± 0.28	0.32 ± 0.81	−0.04 ± 0.37	−0.11 ± 0.02	−1.01 ± 0.58
	Arg384	−2.70 ± 0.60	−2.94 ± 1.58	2.21 ± 0.94	−0.34 ± 0.05	−3.78 ± 1.19
QeH-(*S*)-QE	Tyr189	−1.52 ± 0.47	−1.14 ± 0.88	1.69 ± 0.91	−0.15 ± 0.05	−1.13 ± 0.51
	Phe326	−1.37 ± 0.51	0.02 ± 0.21	0.29 ± 0.20	−0.20 ± 0.07	−1.26 ± 0.57
	Glu328	−2.63 ± 0.47	−3.46 ± 1.29	4.62 ± 1.48	−0.27 ± 0.06	−1.75 ± 0.64
	Tyr350	−1.45 ± 0.42	−0.15 ± 0.16	0.48 ± 0.15	−0.15 ± 0.04	−1.27 ± 0.42
	Trp351	−3.24 ± 0.46	−1.39 ± 1.00	0.94 ± 0.19	−0.21 ± 0.05	−3.91 ± 1.17
	Val354	−1.97 ± 0.38	−0.31 ± 0.20	0.66 ± 0.14	−0.31 ± 0.04	−1.94 ± 0.41

**Table 6 ijms-25-09964-t006:** The key receptor–ligand interactions at the active sites of QeH-(*R*)-QE and QeH-(*S*)-QE complexes with equilibrium time scopes of independent MD trajectory (measurement unit: Å).

System	Time Scope (ns)	Residue	Interaction Mode	Mean of Interaction Distance	SD of Interaction Distance
QeH-(*R*)-QE	150–500	Glu328	π-anion	7.54	0.47
		**Tyr331**	**π-π stacked**	5.15	0.51
		Trp351	π-π stacked	5.25	0.42
		**Arg384**	**π-cation**	5.74	0.30
		**Ser78**	**catalytic site**	7.39	0.69
QeH-(*S*)-QE	300–500	Glu328	π-anion	5.84	0.33
		Tyr331	π-alkyl	6.41	1.52
		Trp351	π-π stacked	5.35	0.32
		**Ser78**	**catalytic site**	8.87	1.07

**Table 7 ijms-25-09964-t007:** Computational alanine scanning (CAS) results for the important residues contributing to the binding affinity (measurement unit: kcal·mol^−1^).

**System**	**Mutation**	**Computational Mutation Energy (ΔΔG*_mut_*)**
QeH-(*R*)-QE	Glu328 > ALA	1.27
	Tyr331 > ALA	0.34
	Tyr350 > ALA	0.75
	Trp351 > ALA	2.01
	Gly352 > ALA	1.29
	Arg384 > ALA	0.95
QeH-(*S*)-QE	Tyr189> ALA	1.17
	Phe326 > ALA	0.51
	Glu328 > ALA	0.89
	Tyr350 > ALA	0.9
	Trp351 > ALA	2.02
	Val354 > ALA	0.44

## Data Availability

The original contributions presented in the study are included in the article, further inquiries can be directed to the corresponding authors.

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
