# Peer review of "Mechanistic Insight into the Enantioselective Degradation of Esterase QeH to (R)/(S)–Quizalofop–Ethyl with Molecular Dynamics Simulation Using a Residue-Specific Force Field"

_ijms, 2024, doi:10.3390/ijms25189964_

Round 1

Reviewer 1 Report

Comments and Suggestions for Authors

With the article “Mechanistic Insight into the Enantioselective Degradation of QeH to (R)/(S)-Quizalofop-ethyl with Molecular Dynamics Simulation Using a Residue-Specific Force Field,” Zhu et al. present an in-depth analysis through molecular dynamics of the possible mechanism of enantioselective hydrolysis of a potential pollutant for bioremediation purposes. I find the article interesting and believe it deserves publication after some suggested modifications are made. Specifically, I think the catalytic mechanisms of the esterase in question should be emphasized more. Currently, as the article is written, these aspects can only be inferred from the numerical measurements of docking and dynamics calculations. I recommend highlighting the mechanisms with diagrams, figures, and possibly a dedicated section that provides an immediate understanding of the findings for a reader in a hurry. Below are my detailed comments on the manuscript. Thank you.

Title
First, I have a suggestion regarding the title of the article. The current title mentions “QeH” without specifying that this is an esterase from Sphingobium sp. It should be clarified at least that it is an esterase for the reader’s understanding.

Abstract

I believe the abstract could be streamlined by removing some superfluous sentences that do not add value to this section. Specifically, lines 23-26 could be considered for removal.

Additionally, it is important to note that line 18 of the abstract mentions computational chemistry but also refers to “experimental approaches” which I did not find in the manuscript. The esterase has not been purified nor cloned through vectors to test it in the reaction under analysis. I think that such a test could add significant value to the manuscript, and I encourage the authors to consider adding an actual experimental section as stated in the abstract. I understand that Zhou et al. identified the experimental phenomenon, but this aspect is not addressed in this article.

Line 28 “is closer” and line 30 “less close” are too generic and should be justified with numerical data (how close and how less close). Also, “a lot” in line 32 does not convey adequate scientific terminology. The authors might consider replacing these terms.

Introduction

As mentioned earlier, it is necessary to show (somewhere, preferably at the beginning of the manuscript) the reaction scheme with molecular representations of the substrate (racemate) and the expected hydrolysis by the esterase, showing the resolution of the two different enantiomers.

In line 44, data from 2014 is mentioned. I wonder if more recent data is available. Please update.

Line 83, reference number 17: it would be interesting for the reader to briefly know why the result was considered “unreliable.”

The article often refers to the “newly-developed residue-specific force field (RSFF2C).” However, there is no extended discussion of this aspect, its advantages, differences compared to conventional force fields, or how it was obtained in the materials and methods section. Please clarify this aspect.

Results and Discussion

Line 97 reports the binding energies of the two enantiomers. I wonder if these measurements were done in singlet or triplicate, and if in triplicate, the values could be reported as mean and standard deviation. Additionally, since the biocatalyst used is a serine hydrolase, I wonder what the distances are between the oxygen of the serine in the catalytic triad and the carbonyl carbon of the ester to be attacked (as reported in these previous works: 1. https://doi.org/10.3390/app13179852, 2. https://doi.org/10.1002/cssc.202102657). Please comment based on the provided literature.

Line 209: “to decipher the enantioselective degradation mechanism of (R/S)-QE catalyzed by QeH” could be the point to actually summarize and synthesize the entire reasoning regarding the hydrolysis mechanism that overall has been identified in this article.

Materials and Methods

Section 3.1: To make the article reproducible, please provide a step-by-step procedure that allows the reader, through Alphafold, to obtain the exact same esterase model used by the authors for their tests.

Section 3.2: The coordinates for the center of the docking box are provided, but it is not indicated how (with what parameters) these were defined. It would be helpful to include this information.

English Form

I noticed some minor formatting issues that I ask the authors to consider and correct:

Line 19: “the mechanism of QeH how to adjust its conformation to adapt to substrate binding” – please correct this sentence as it is difficult to read.

Line 52: “Quizalofop-ethyl (QE), one of the wildly used chiral AOPP herbicides.” I believe “is” is missing before “one” since the sentence ends shortly after.

Please review the entire article for minor typos. Thank you.

Comments on the Quality of English Language

Minor editing of English language required

Author Response

Response to comments point-by-point

Reviewer #1:

With the article “Mechanistic Insight into the Enantioselective Degradation of QeH to (R)/(S)-Quizalofop-ethyl with Molecular Dynamics Simulation Using a Residue-Specific Force Field,” Zhu et al. present an in-depth analysis through molecular dynamics of the possible mechanism of enantioselective hydrolysis of a potential pollutant for bioremediation purposes. I find the article interesting and believe it deserves publication after some suggested modifications are made. Specifically, I think the catalytic mechanisms of the esterase in question should be emphasized more. Currently, as the article is written, these aspects can only be inferred from the numerical measurements of docking and dynamics calculations. I recommend highlighting the mechanisms with diagrams, figures, and possibly a dedicated section that provides an immediate understanding of the findings for a reader in a hurry. Below are my detailed comments on the manuscript. Thank you.

Comments 1. Title: First, I have a suggestion regarding the title of the article. The current title mentions “QeH” without specifying that this is an esterase from Sphingobium sp. It should be clarified at least that it is an esterase for the reader’s understanding.

Response: Thanks for your good suggestion. The title of this paper has been revised as “Mechanistic Insight into the Enantioselective Degradation of Esterase QeH to (R)/(S)-Quizalofop-ethyl with Molecular Dynamics Simulation Using a Residue-Specific Force Field”.

Comments 2. Abstract: I believe the abstract could be streamlined by removing some superfluous sentences that do not add value to this section. Specifically, lines 23-26 could be considered for removal.

Response: Yes, thanks for your suggestion and we have removed these superfluous sentences.

Comments 3. Additionally, it is important to note that line 18 of the abstract mentions computational chemistry but also refers to “experimental approaches” which I did not find in the manuscript. The esterase has not been purified nor cloned through vectors to test it in the reaction under analysis. I think that such a test could add significant value to the manuscript, and I encourage the authors to consider adding an actual experimental section as stated in the abstract. I understand that Zhou et al. identified the experimental phenomenon, but this aspect is not addressed in this article.

Response: Thanks for your positive and constructive comments and suggestions on our manuscript. The His-tagged recombinant QeH has been purified after heterologous expression of positive recombinant plasmids (pET-qeH) in Escherichia coli BL21 (DE3), and the biochemical characterization and substrate specificity of QeH have been studied systematically (Zhou et al., 2020). QeH preferentially hydrolyzed (R)-quizalofop-ethyl than (S)-quizalofop-ethyl. Based on molecular docking and molecular dynamics analyses, several key amino acid sites that affecting the catalytic activity of QeH were identified. Next, our work will carry out experiments on the above findings, carry out site-directed mutation on wild-type QeH, verify our theoretical calculation conclusions, and directionally reform and design the enzyme to improve its catalytic efficiency.

Reference

Zhou X., Zhang L., Wei L., Cai J., Chen K., Jiang J. Characterization of an enantioselective esterase from the quizalofop-ethyl-transforming strain of Sphingobium sp. QE-1. International Biodeterioration & Biodegradation 2020, 155, 105104, doi:10.1016/j.ibiod.2020.105104.

Comments 4. Line 28 “is closer” and line 30 “less close” are too generic and should be justified with numerical data (how close and how less close). Also, “a lot” in line 32 does not convey adequate scientific terminology. The authors might consider replacing these terms.

Response: Thank you for your good suggestion. We have modified it as “Our results indicated that the benzene ring of the (R)-QE structure can simultaneously form anion-π and cation-π interactions with the side-chain group of Glu328 and Arg384 in the binding cavity of QeH-(R)-QE complex, resulting that (R)-QE is closer to the catalytic triplet system (Ser78-Lys81-Tyr189) with the distances measured for the hydroxyl oxygen atom of the catalytic Ser78 of QeH and the carbonyl carbon atom of (R)-QE of 7.39 Å, compared to the 8.87 Å for (S)-QE;” in the revised abstract section. In addition, The time-dependent distances between the hydroxyl oxygen atom of the catalytic Ser78 of QeH and the carbonyl carbon atom of (R)/(S)-QE were measured and displayed in Figure 7A. The representation of the catalytic triad of QeH and the substrate (R)/(S)-QE was also given in Figure 7B and C. The corresponding results have been discussed in the revised Results and discussion section.

Comments 5. Introduction: As mentioned earlier, it is necessary to show (somewhere, preferably at the beginning of the manuscript) the reaction scheme with molecular representations of the substrate (racemate) and the expected hydrolysis by the esterase, showing the resolution of the two different enantiomers.

Response: Thanks for your good suggestion and the proposed catabolic pathway of (R)-quizalofop-ethyl and (S)-quizalofop-ethyl by the esterase QeH is showing the bellow figure. QeH preferentially hydrolyzed (R)-quizalofop-ethyl than (S)-quizalofop-ethyl.

Figure 1 The proposed catabolic pathway of (R)-quizalofop-ethyl and (S)-quizalofop-ethyl by the esterase QeH

Comments 6. In line 44, data from 2014 is mentioned. I wonder if more recent data is available. Please update.

Response: Thanks for your comments. We have searched numerous references, but failed to found more recent data of the application amount or global sales of AOPP herbicides. Still and all, the problem of AOPP herbicide residues in the environment cannot be ignored. Many studies have focused on estimating the potential toxicity of the chiral AOPP herbicide residues. For example, the biotoxic response of target protein to chiral diclofop-methyl has significant enantioselectivity. Toxic affinity of (S)-enantiomer for target protein ~1.5 times that of (R)-enantiomer (Ding et al., 2020). Exposure of zebrafish embryos to 0.1, 0.3 and 0.5 mg L-1 diclofop-methyl from 6 hours post fertilization (hpf) to 72 hpf induced developmental abnormalities, such as shorter body lengths and yolk sac edemas (Cao et al., 2019). Exposure of zebrafish embryos to 0.75, 1.0 and 1.25 mg L-1 diclofop-methyl induced cardiac defects, such as pericardial edema, slow heart rate and long SV-BA distance (Cao et al., 2020). Metamifop exhibited high acute toxicity to zebrafish, with 96 h-LC50 values of 0.648 and 0.216 mg L-1 to embryos and larvae of 72 h post-hatching (hph), respectively (Zhao et al., 2019). Quizalofop-P-ethyl exposure significantly increased the mortality rate, decreased the hatching rate and caused morphological defects during zebrafish embryonic development (Zhu et al., 2022). However, microbial catabolism is considered the major pathway for the dissipation of AOPP herbicides in the environment. (Huang et al., 2017; Zhou et al., 2018). Because of cost-effectiveness and environmental compatibility, bioremediation using microorganisms has an excellent potential for future development (Saeed et al., 2022). Therefore, clarifying their catabolic molecular mechanism will provide strong support for the bioremediation of polluted environment.

References

  1. Cao, Z.G., Huang, Y., Xiao, J.H., Cao, H., Peng, Y.Y., Chen, Z.Y., et al., 2020. Exposure to diclofop-methyl induces cardiac developmental toxicity in zebrafish embryos. Environ Pollut 259, 113926. https://doi.org/1016/j.envpol.2020.113926.
  2. Cao, Z.G., Zou, L.F., Wang, H.L., Zhang, H., Liao, X.J., Xiao, J.H., et al., 2019. Exposure to diclofop-methyl induces immunotoxicity and behavioral abnormalities in zebrafish embryos. Aquat Toxicol 214, 105253. https://doi.org/1016/j.aquatox.2019.105253.
  3. Ding, F., Peng, W., Peng, Y.K., Liu, B.Q., 2020. Estimating the potential toxicity of chiral diclofop-methyl: Mechanistic insight into the enantioselective behavior. Toxicology 438, 152446. https://doi.org/10.1016/j.tox.2020.152446.
  4. Huang, X., He, J., Yan, X., Hong, Q., Chen, K., He, Q., et al., 2017. Microbial catabolism of chemical herbicides: Microbial resources, metabolic pathways and catabolic genes. Pestic Biochem Physiol 143, 272–297. https://doi.org/10.1016/j.pestbp.2016.11.010.
  5. Saeed, M.U., Hussain, N., Sumrin, A., Shahbaz, A., Noor, S., Bilal, M., et al., 2022. Microbial bioremediation strategies with wastewater treatment potentialities-A review. Sci Total Environ 818, 151754. https://doi.org/10.1016/j.scitotenv.2021.151754.
  6. Zhao, F., Li, H., Cao, F., Chen, X., Liang, Y., Qiu, L., 2019. Short-term developmental toxicity and potential mechanisms of the herbicide metamifop to zebrafish (Danio rerio) embryos. Chemosphere 236, 124590. https://doi.org/10.1016/j.chemosphere.2019.124590.
  7. Zhou, J., Liu, K., Xin, F.X., Ma, J.F., Xu, N., Zhang, W.M., et al., 2018. Recent insights into the microbial catabolism of aryloxyphenoxy-propionate herbicides: microbial resources metabolic pathways and catabolic enzymes. World J Microbiol Biotechnol 34, 117. https://doi.org/10.1007/s11274-018-2503-y.
  8. Zhu, L., Wang, C., Jiang, H., Zhang, L., Mao, L., Zhang, Y., et al., 2022. Quizalofop-P-ethyl induced developmental toxicity and cardiotoxicity in early life stage of zebrafish (Danio rerio). Ecotoxicol Environ Saf 238, 113596. https://doi.org/10.1016/j.ecoenv.2022.113596.

Comments 7. Line 83, reference number 17: it would be interesting for the reader to briefly know why the result was considered “unreliable.”

Response: Thanks for your good comments. We have added it as “Molecular docking can only obtain the static interaction model of enzyme-catalyzed complex. Since this approach takes no account of the structural dynamics and has no guarantee of global minimum. For the PsaA system, its swep-bound complex is extremely unstable in its docking pose during our recent MD study, and it undergoes two-stage fluctuations toward its unbounded state due to unfavorable interactions between both of them. How QeH adjusts its conformation and adapts to substrate binding as well as facilitates protein-ligand interactions still have to be further simulated by performing MD simulations [17].” in the revised Introduction section.

Comments 8. The article often refers to the “newly-developed residue-specific force field (RSFF2C).” However, there is no extended discussion of this aspect, its advantages, differences compared to conventional force fields, or how it was obtained in the materials and methods section. Please clarify this aspect.

Response: Thanks for your good comments. In the revised Introduction section, we have added its development and advantages as “The MD reliability depended on the accuracy with which the interactions between atoms are calculated [17]. Based on the Protein Data Bank (PDB), the pre-residue local conformational preference (coil library) was firstly constructed by Wu et al. [18]. Through statistical analysis of coil library, it was found that there were significant differences between the intrinsic conformational preferences of residues and the conformational distributions (backbone dihedral angles φ, ψ and side-chain torsion angles χi) given by conventional force fields. The optimization and correction of pre-residue conformational preferences were achieved by optimizing the dihedral parameters and introducing special local non-bonded interactions [18-19]. Therefore, Wu et al. [18-20] designed and developed a range of residue-specific force fields (RSFFs) that can describe binding dynamics, thermodynamics and intermolecular forces more accurately at the atomic level. In this paper, a newly-developed residue-specific force field (RSFF2C) with high accuracy and efficiency was adopted to simulate the enantioselective catalytic dynamics process and thermodynamic properties of QeH against the different enantiomers of QE”.

In the revised Materials and Methods section, we have also provided the specific operation steps of RSFF2C in AMBER software as “The force field setup began by generating the initial topology and coordinate files in AmberTools using the 'tleap' module based on the AMBER ff14SB force field. The generated topology file was modified by adding CMAP correction terms and other dihedral angle corrections to implement the RSFF2C force field [18-20]. The modified topology file was used for subsequent molecular dynamics simulations. Through these steps, the topology and parameter files for the protein-ligand complex compatible with the RSFF2C+GAFF force fields were successfully generated, providing a reliable foundation for molecular dynamics simulations.”

Comments 9. Line 97 reports the binding energies of the two enantiomers. I wonder if these measurements were done in singlet or triplicate, and if in triplicate, the values could be reported as mean and standard deviation. Additionally, since the biocatalyst used is a serine hydrolase, I wonder what the distances are between the oxygen of the serine in the catalytic triad and the carbonyl carbon of the ester to be attacked (as reported in these previous works: 1. https://doi.org/10.3390/app13179852, 2. https://doi.org/10.1002/cssc.202102657). Please comment based on the provided literature.

Response: Thanks for your good comments. The binding energies of ten independent QeH and (R)/(S)-QE complexes were summarized in Table 2, and their mean and standard deviation has also been listed in the table.

Ligand

Structure

Docking Energy

Mean

Standard Deviation

(R)-QE

1

-9.26

-7.55

1.87

2

-9.17

3

-8.93

4

-8.87

5

-8.65

6

-8.54

7

-6.74

8

-6.18

9

-4.58

10

-4.57

(S)-QE

1

-8.79

-6.98

0.98

2

-8.29

3

-7.09

4

-7.04

5

-6.91

6

-6.89

7

-6.78

8

-6.67

9

-5.73

10

-5.60

Table 2. The docking results of esterase QeH to (R)/(S)-QE. (Measurement Unit: kcal·mol-1)

Thank you for these two previous articles you provided, and the time-dependent distances between the hydroxyl oxygen atom of the catalytic Ser78 of QeH and the carbonyl carbon atom of (R)/(S)-QE were also measured and displayed in Figure 7A and Table 6. The representation of the catalytic triad of QeH and the substrate (R)/(S)-QE was also given in Figure 7B and C. The corresponding results have been discussed in the revised Results and discussion section as “During the MD simulation process, the cation-π and anion-π interactions simultaneously formed between the benzene ring of (R)-QE and the side chains of Glu328 and Arg384, resulting that (R)-QE was closer to the catalytic triplet system (Ser78-Lys81-Tyr189) with the distances measured for the hydroxyl oxygen atom of the catalytic Ser78 of QeH and the carbonyl carbon atom of (R)-QE of 7.39 Å, compared to the 8.87 Å for (S)-QE (Figure 7 and Table 6) [16]”.

Figure 7. Time-dependent distances between the hydroxyl oxygen atom of the catalytic Ser78 of QeH and the carbonyl carbon atom of (R)/(S)-QE (A), and the representation of the catalytic triad of QeH and the substrate (R)/(S)-QE (B and C). The catalytic triad (Ser78, Lys81, and Tyr189) of QeH is in blue and green, whereas the (R)/(S)-QE is salmon.

Table 6. The key receptor-ligand interactions at the active sites of QeH-(R)-QE and QeH-(S)-QE complexes with equilibrium time scopes of independent MD trajectory. (Measurement Unit: Å)

System

Time scope (ns)

Residue

Interaction mode

Mean of interaction distance

SD of interaction distance

QeH-(R)-QE

150-500

Glu328

π-anion

7.54

0.47

Tyr331

π-π stacked

5.15

0.51

Trp351

π-π stacked

5.25

0.42

Arg384

π-cation

5.74

0.30

Ser78

catalytic site

7.39

0.69

QeH-(S)-QE

300-500

Glu328

π-anion

5.84

0.33

Tyr331

π-alkyl

6.41

1.52

Trp351

π-π stacked

5.35

0.32

Ser78

catalytic site

8.87

1.07

Comments 10. Line 209: “to decipher the enantioselective degradation mechanism of (R/S)-QE catalyzed by QeH” could be the point to actually summarize and synthesize the entire reasoning regarding the hydrolysis mechanism that overall has been identified in this article.

Response: Thanks for your good comments. The ester bond hydrolysis process of (R)-QE catalyzed by esterase QeH was shown schematically in Figure 8.

Figure 8. Schematic illustration of the ester bond hydrolysis process of (R)-QE catalyzed by esterase QeH.

The hydrolysis mechanism was summarized in the revised 2.3 section as “As shown in Figure 8, the ester bond hydrolysis of (R)-QE catalyzed by esterase QeH can be decomposed in to three steps. The anion-π, cation-π and π-π stacking interactions involved by identified key amino acids of QeH facilitate (R)-QE to bind to the catalytic Ser78, and its hydroxyl hydrogen atom is transferred to Lys81 to form an oxygen anion and attack the carbonyl carbon atom of (R)-QE to form a C-O bond. Then the hydrogen atom of Tyr189 is transferred to the ester oxygen atom of (R)-QE, and a hydrogen atom from Lys81 simultaneously removes an ethanol molecule for Tyr189. Finally, the ester bond formed by (R)-QE and Ser78 is hydrolyzed under the action of water molecules, and the catalytic active site is reduced to the initial state. (R)-QE is hydrolyzed into two parts: carboxyl group and hydroxyl group.”.

Comments 11. Materials and Methods: Section 3.1: To make the article reproducible, please provide a step-by-step procedure that allows the reader, through Alphafold, to obtain the exact same esterase model used by the authors for their tests.

Response: Thanks for your good comments. The step-by-step procedure of predicting the esterase model by AlphaFold2 were added as “Since the protein crystal structure of the esterase QeH was not available in the Protein Data Bank (PDB) at present, AlphaFold2 was used to predict the three-dimensional structure of QeH [28]. The structural prediction in this study was conducted on a Linux operating system using a Python 3.9.7 and CUDA 10.2.89 environment, with the configuration of AlphaFold2 simplified through Docker scripts. Several databases required by AlphaFold2, including BFD, MGnify, PDB70, PDB, PDB seqres, UniRef30, UniProt, and UniRef90, were pre-downloaded and stored in a designated directory. The target protein esterase QeH’s sequence in FASTA format was prepared and named “QeH.fasta”. AlphaFold2 was then run for structural prediction using the following command: 1) source activate; 2) conda activate alphafold2; 3) python /opt/alphafold2/docker/run_docker.py --fasta_paths=QeH.fasta --max_template_date=2022-07-06 --db_preset='full_dbs' --data_dir=/home/hipeson/software/MSA --gpu_devices 0. In this command: fasta_paths: Specifies the path to the FASTA file containing the target protein sequence, in this case, “QeH.fasta”; max_template_date: Sets the maximum template date to ensure that the template data used is no later than this date; model_preset: Selects the model configuration; “full_dbs” indicates that all genetic databases from CASP14 are used; data_dir: Specifies the directory where the databases are stored, here “/home/hipeson/software/MSA”; gpu_devices: Specifies the GPU device to be used for computation, with “0” indicating the first GPU. The predicted ranker_0 structure with the highest score of 94.696 (Table 1) was selected as the structure model of QeH for subsequent calculations [29].”

Comments 12. Section 3.2: The coordinates for the center of the docking box are provided, but it is not indicated how (with what parameters) these were defined. It would be helpful to include this information.

Response: Thanks for your good comments. We have revised it as “The coordinates of the center of the docking box were (1.03, 1.29, 0.08) based on the active site located by catalytic triads (Ser78-Lys81-Tyr189)”.

Comments 13. Line 19: “the mechanism of QeH how to adjust its conformation to adapt to substrate binding” – please correct this sentence as it is difficult to read.

Response: Thanks for your suggestion. I'm sorry for writing these poor sentences, we have revised it as “it is still unclear how the esterase QeH to adjusts its conformation to adapt to substrate binding and catalysis and promote enzyme-substrate interaction in the catalytic kinetics.”. We also checked throughout the manuscript and other inappropriate sentences have been polished.

Comments 14. Line 52: “Quizalofop-ethyl (QE), one of the wildly used chiral AOPP herbicides.” I believe “is” is missing before “one” since the sentence ends shortly after. Please review the entire article for minor typos

Response: Thanks for your suggestion. We have revised it as “Quizalofop-ethyl (QE) is one of the wildly used chiral AOPP herbicides”. We have checked and corrected all these errors in the article.

Reviewer 2 Report

Comments and Suggestions for Authors

The manuscript presents an interesting approach with Docking and MD to understanding a complex biochemical process. However, I have some minor revision suggestions to improve the clarity and comprehensiveness of the manuscript:

1. The resolution of Figure 4 for the interaction diagram is not sufficient for readers to clearly understand the details. Please consider increasing the figure resolution or providing a zoomed-in version of the plot.

2. While the authors mention using the AMBER force field and residue-specific force field, they do not provide sufficient information on both the protein and ligand components. For example is that similar to AMBER99+ GAFF2 ? What was the atomic charge model ?

3. The authors use the PM3 charge model in the docking step. Could they please explain why this particular charge model?

4. The author should also present the energy ranking of different poses from the docking results in the SI.

5. Please include unit labels for all tables (e.g., "kcal/mol") in the captions of each table.

Author Response

Response to comments point-by-point

Reviewer #2

The manuscript presents an interesting approach with Docking and MD to understanding a complex biochemical process. However, I have some minor revision suggestions to improve the clarity and comprehensiveness of the manuscript:

Comments 1. The resolution of Figure 4 for the interaction diagram is not sufficient for readers to clearly understand the details. Please consider increasing the figure resolution or providing a zoomed-in version of the plot.

Response: Thanks for your suggestions. This figure has been redrawn in the revised manuscript.

Figure 6. The key interactions at the active sites of the representative conformations of the QeH-(R)-QE and QeH-(S)-QE complexes with equilibrium stabilization. The interactions derived from the representative conformation of the QeH-(R)-QE (A and B) and QeH-(S)-QE (C and D) complexes generated by the MD simulations were represented by dotted lines in different colors, and the unit of interaction distances was Å.

Comments 2. While the authors mention using the AMBER force field and residue-specific force field, they do not provide sufficient information on both the protein and ligand components. For example is that similar to AMBER99+ GAFF2 ? What was the atomic charge model?

Response: Thanks for your good suggestions. The details about using AMBER software to process the protein and ligand have been provided in the revised 3.3 section as follows: “The QeH structure was checked using Discovery Studio (DS) 2019 software and all hydrogen atoms were removed [37]. The substrates of (R)/(S)-QE were preprocessed using the GAFF force field. Its electrostatic potential (ESP) charges were calculated at the HF/6-31G* level using Gaussian16 [38], and the computed electron density fitted to the atomic charges by using the restrained electrostatic potential (RESP) procedure of Antechamber. Subsequently, the coordinates of (R)/(S)-QE were copied to the structure of QeH to form a complex. The force field setup began by generating the initial topology and coordinate files in AmberTools using the 'tleap' module based on the AMBER ff14SB force field. The generated topology file was modified by adding CMAP correction terms and other dihedral angle corrections to implement the RSFF2C force field [18-20]. The modified topology file was used for subsequent molecular dynamics simulations. Through these steps, the topology and parameter files for the protein-ligand complex compatible with the RSFF2C+GAFF force fields were successfully generated, providing a reliable foundation for molecular dynamics simulations.”

Comments 3. The authors use the PM3 charge model in the docking step. Could they please explain why this particular charge model?

Response: Thanks for your comments. The combination of quantum mechanics (QM) and molecular mechanics (MM) is a common approach for investigating the protein-ligand interactions and enzyme reactions. The active site or binding site is usually treated by the ab initio density functional theory or semi-empirical potentials, whereas the rest of the system is calculated by the force fields based on molecular mechanics during molecular docking process. In the current version of sander, one can use the PM3 semi-empirical Hamiltonian for the quantum mechanical region. Interaction between the QM and MM regions includes electrostatics (based on partial charges in the MM part) and Lennard–Jones terms, designed to mimic the exchange-repulsion terms that keep QM and MM atoms from overlapping. The primary reason for choosing the PM3 charge model is its computational efficiency and suitability within semi-empirical quantum chemistry methods (Yilmaz H et al., 2016). The PM3 charge model can provide a reasonable charge distribution in a shorter amount of time, which is particularly important when dealing with the large molecular systems or when multiple simulations are required. Additionally, the PM3 model performs well in predicting the geometry and electronic properties of many biomolecules and organic molecules, making it widely applicable in molecular docking and molecular dynamics simulations (Al Azzam K. M et al., 2015).

Reference

Yilmaz H., Ahmed L., Rasulev B., Leszczynski J. Application of ligand-and receptor-based approaches for prediction of the HIV-RT inhibitory activity of fullerene derivatives. Journal of Nanoparticle Research 2016, 18(5), 123, doi: 10.1007/s11051-016-3429-7.

Al Azzam K. M., Muhammad E. Host-guest Inclusion Complexes between Mitiglinide and the Naturally Occurring Cyclodextrins α, β, and γ: A Theoretical Approach. Advanced pharmaceutical bulletin 2015, 5(2), 289-291, doi:10.15171/apb.2015.040.

Comments 4.  The author should also present the energy ranking of different poses from the docking results in the SI.

Response: Thanks for your suggestions. The energy ranking of different poses from the docking results were summarized in Table 2, and their mean and standard deviation has also been listed in this table.

In addition, in the revised Results and Discussions section, we have modified as “The docking results for the ten independent QeH and (R)/(S)-QE complexes were summarized in Table 2. As shown in Figure R2-1, conformational cluster analysis of the docked complexes showed that their binding poses belonged the same type of conformation and the ten representative snapshots of (R)/(S)-QE superposed inside the interior of their respective hydrophobic cavity. It was displayed that the binding energies of esterase QeH to (R)-QE and (S)-QE corresponding to the preferred pose were calculated to be -9.26 and -8.79 kcal·mol-1, respectively.”.

Table 2. The docking results of esterase QeH to (R)/(S)-QE. (Measurement Unit: kcal·mol-1)

Ligand

Structure

Docking Energy

Mean

Standard Deviation

(R)-QE

1

-9.26

-7.55

1.87

2

-9.17

3

-8.93

4

-8.87

5

-8.65

6

-8.54

7

-6.74

8

-6.18

9

-4.58

10

-4.57

(S)-QE

1

-8.79

-6.98

0.98

2

-8.29

3

-7.09

4

-7.04

5

-6.91

6

-6.89

7

-6.78

8

-6.67

9

-5.73

10

-5.60

Figure R2-1 The ten representative snapshots of (R)-QE (A) and (S)-QE (B) superposed at their respective QeH binding sites inside the interior of hydrophobic pocket obtained from the docket results.

Comments 5.  Please include unit labels for all tables (e.g., "kcal/mol") in the captions of each table.

Response: Thanks for your suggestions. All the tables have been included the “Measurement Unit: kcal·mol-1” or “Measurement Unit: kcal·Å” in the revised captions.

Reviewer 3 Report

Comments and Suggestions for Authors

Please see attached file for detailed comments.

Comments on the Quality of English Language

The English is a little unclear in some places in the manuscript but it is not major. If possible, having a native speaker look over the manuscript would be helpful to make everything clear.

Author Response

Response to comments point-by-point

Reviewer #3:

In this work, Zhu et al. explore the enantioselective degradation dynamics of quizalofop-ethyl (QE) enantiomers by the QeH esterase. The study is a computational effort, making use of a recently published residue-specific force field RSFF2C to conduct molecular dynamics simulations to determine equilibrium complexes between QeH and both (R)-QE and (S)-QE enantiomers, from which binding free energies and the relative contributing factors to these binding free energies are computed. These simulations lead the authors to conclude that QeH esterase preferentially binds to and catalytically degrades the (R)-QE enantiomer. Further residue analyses reveal that the (R)-QE enantiomer has two potential interaction sources that contribute to the formation of a stable complex with QeH, namely the  stacking interactions between certain residues on QeH esterase and (R)-QE’s phenyl ring moieties as well as anionic-  and cationic-  interactions between this same phenyl ring and other side chains on the estersase. On the other hand, (S)-QE only has  stacking between it’s phenyl ring and one residue on the esterase, and lacks the cationic-  interactions of it’s enantiomer, leading to the conclusion that these weaker interactions explain why QeH esterase preferentially degrades the R enantiomer. Finally, computational alanine scanning studies highlight the importance of the  stacking interactions between the residue (specifically Trp351) and the phenyl ring and that this needs to be maintained to ensure catalytic activity. Overall, the methodology and findings of the paper are interesting, but there are technical and formatting concerns that should be addressed before the paper is published.

Comments 1: The authors use AlphaFold2 to generate the 3D structure for QeH as no crystal structure is available in the Protein Data Bank. They took the best scoring structure and said that this has high reliability but do not comment further. How many structures did AlphaFold2 generate for QeH and how much better was the ranker_0 structure than the next best structure? Deep learning models are not perfect, so often a few candidate structures would be considered instead of just what the model predicts as the best one. Can the authors comment on this?

Response: Thank you for your comment. We appreciate your concerns about the model selection process and the evaluation of multiple candidate structures. In the revised Results and Discussions section, we have added it as “The structure models for QeH predicted by Alphafold2 was evaluated using the PROCHECK and Verify 3D programs, with the results shown in Figure 2. The Ramachandran plot illustrates the allowed and disallowed conformations of amino acid residues in proteins, where the red regions indicate the most favoured areas, the yellow regions represent the generously allowed areas, and the blank regions is the disallowed areas. As shown in Figure 2A, all the residues of QeH model are absent from disallowed regions, consistent with the stereochemical energy rules. The Verify 3D program assessed the compatibility of the protein’s 3D model with its amino acid sequence. As shown in Figure 2B, 99.54% of the amino acid residues scored above 0.1, meeting the evaluation criteria. Combining these analyses with the high confidence (predicted local distance difference test, pLDDT) scores from AlphaFold2 (Table 1), the predicted Rank_0 structure is confirmed to be reasonable and suitable for further calculations.”

Table 1. The pLDDT Scores of structure predictions using AlphaFold2.

Esterase

Structure

Rank Score

QeH

Rank_0

94.696

Rank_1

94.675

Rank_2

94.540

Rank_3

94.322

Rank_4

94.261

Figure 2. Ramachandran plot (A) and verify 3D score (B) for the predicted model of QeH.

The details about the predicting structures by AlphaFold2 have been added in the Materials and Methods section as “Since the protein crystal structure of the esterase QeH was not available in the Protein Data Bank (PDB) at present, AlphaFold2 was used to predict the three-dimensional structure of QeH [28]. The structural prediction in this study was conducted on a Linux operating system using a Python 3.9.7 and CUDA 10.2.89 environment, with the configuration of AlphaFold2 simplified through Docker scripts. Several databases required by AlphaFold2, including BFD, MGnify, PDB70, PDB, PDB seqres, UniRef30, UniProt, and UniRef90, were pre-downloaded and stored in a designated directory. The target protein esterase QeH’s sequence in FASTA format was prepared and named “QeH.fasta”. AlphaFold2 was then run for structural prediction using the following command.” and “The predicted ranker_0 structure with the highest score of 94.696 (Table 1) was selected as the structure model of QeH for subsequent calculations”.

Comments 2: Additionally, did the authors compare the generated AlphaFold2 3D structure for QeH to crystal structure of similar proteins? Or are there no similar proteins to compare to? The authors should comment more on this.

Response: Thank you for your comment. The QeH sequence was subjected to a BLAST search in the PDB protein database. Based on the search results, an enzyme (PDB ID: 7PP3) with the highest similarity to the QeH sequence was selected, and they exhibit 46.37% sequence identity. As shown in Figure R1, the predicted structure of QeH was superimposed with its structure, and the backbone RMSD value was 0.791 Å.

Figure R1. Superimposition of the predicted structure of QeH (blue) with the crystal structure of an enzyme (PDB ID: 7PP3) (green).

Comments 3: For the docking simulations the authors conduct 10 docking runs in AutoDock. This seems fine, but the authors do not report how many poses were generated by each run. Could the authors report this to give an idea of how much pose filtering was necessary? Also, the authors said they select the optimal poses as input to the MD simulations. To clarify, does this mean that there are 10 initial guesses for each complex? Like with the above, if only one pose per run was retained, could the authors provide details on how much more preferable this optimal pose was?

Response: Thank you for your comment. The energy ranking of different poses from the docking results were summarized in Table 2, and their mean and standard deviation has also been listed in this table.

Table 2. The docking results of esterase QeH to (R)/(S)-QE. (Measurement Unit: kcal·mol-1)

Ligand

Structure

Docking Energy

Mean

Standard Deviation

(R)-QE

1

-9.26

-7.55

1.87

2

-9.17

3

-8.93

4

-8.87

5

-8.65

6

-8.54

7

-6.74

8

-6.18

9

-4.58

10

-4.57

(S)-QE

1

-8.79

-6.98

0.98

2

-8.29

3

-7.09

4

-7.04

5

-6.91

6

-6.89

7

-6.78

8

-6.67

9

-5.73

10

-5.60

Figure R2 The ten representative snapshots of (R)-QE (A) and (S)-QE (B) superposed at their respective QeH binding sites inside the interior of hydrophobic pocket obtained from the docket results.

In addition, in the revised Results and Discussions section, we have modified as “The docking results for the ten independent QeH and (R)/(S)-QE complexes were summarized in Table 2. As shown in Figure R2, conformational cluster analysis of the docked complexes showed that their binding poses belonged the same type of conformation and the ten representative snapshots of (R)/(S)-QE superposed inside the interior of their respective hydrophobic cavity. It was displayed that the binding energies of esterase QeH to (R)-QE and (S)-QE corresponding to the preferred pose were calculated to be -9.26 and -8.79 kcal·mol-1, respectively.”.

Comments 4: For the molecular dynamics simulations, could the authors explain why they conducted NVT and NPT simulations successively? Was this to allow further relaxation of the system’s initial configuration after the optimization algorithm before exploring the free energy profiles?

Response: Thank you for your insightful comment regarding our molecular dynamics (MD) simulation protocol. The sequential execution of NVT (constant number of particles, volume, and temperature) and NPT (constant number of particles, pressure, and temperature) simulations was indeed a deliberate choice. This approach is commonly adopted to ensure proper equilibration and relaxation of the system before proceeding with the production run and subsequent analyses.

NVT Simulation: Initially, an NVT simulation was conducted to stabilize the system's temperature while maintaining the volume constant. This step is crucial after the initial energy minimization as it allows the system to reach a desired thermal equilibrium. During this phase, the system’s kinetic energy distribution becomes consistent with the set temperature, ensuring that the temperature fluctuations are under control before adjusting the pressure. NPT Simulation: Following the NVT simulation, the system was transitioned to an NPT ensemble, where the pressure and temperature were maintained constant while allowing the volume to fluctuate. This step ensures that the system reaches the correct density and mimics experimental conditions where the system can respond to pressure changes. The NPT simulation allows the system to relax further and achieve a stable density, ensuring that the subsequent production run (typically under NPT conditions) accurately reflects a physically realistic environment. The combination of these two phases ensures that the system is adequately equilibrated before proceeding to explore the free energy profiles or any further analyses, reducing the risk of artifacts that might arise from an improperly equilibrated system.

Comments 5: In the binding free energy calculation section, the author’s rational for neglecting the entropic contribution is worded a little confusingly. Is the idea that since the property of interest is relative binding free energy between the two enantiomers, it is expected that the entropic contributions will largely cancel out between the two, so there is no need to calculate the individual ones? Can the authors clarify?    

Response: Thank you for your comment. Thank you for your comment. I apologize for the unclear wording regarding the neglect of the entropic contribution. During catalysis, enzymes may undergo conformational changes that can lead to changes in their structural entropy. The entropic effect (-T∆S) should be considered using the normal mode approximation through the NMODE module. We will use this method to recalculate the binding energy.

In the initial draft, we used AMBER's built-in MMPBSA tool to calculate the binding free energy. Similar to GROMACS's `g_mmpbsa` tool, these calculations neglected the entropic term. In our study, as we focused on the relative binding free energy between the two enantiomers, we anticipated that the entropic contributions would largely cancel out between the two, thus we chose to neglect the individual entropic calculations. We have clarified this point in the manuscript, with the revised text as follows: "Given that our study focuses on the relative binding free energy between the two enantiomers, the entropic contribution (-T∆S) is expected to largely cancel out between the two, so it was approximately neglected [45]."

We again thank you for your thorough review and suggestions, and we hope that this revision makes the content clearer. Should there be any further questions or suggestions, we are more than happy to discuss and improve further.

Comments 6: One other main concern is the free energies from which the authors use to justify their conclusions. In general, all the reported free energies for the two enantiomers effectively overlap when considering the author’s reported error bars for their simulations. Reporting of error bars from replicates should be commended, but it does make it difficult to distinguish between the R and S enantiomers of QE. For example, the total binding energies of the QeH-R-QE complex and QeH-S-QE complex are -40.94  3.78 and -38.95  3.78 kcal/mol. Are these statistically significant differences to conclude that QeH esterase preferentially binds to the R enantiomer of QE? Also in this case, why are the error bars identical? The only components that seem statistically different are the strong polar solvation energy and the electrostatic interactions, which the authors do highlight. I think a discussion or statistical test to argue that these differences are statistically significant and beyond the expected error of the molecular dynamics simulation is warranted.

Response: Thank you for your comment. The fractional part of error bars of the total binding energies is identical due to rounds up the law to leave. We have corrected it as “-40.94 ± 3.7851 and -38.95 ± 3.7882 kcal·mol-1” in the revised manuscript. In addition, statistical analysis is required as suggested by the reviewer, we calculated the time-dependent binding energies of the QeH-(R)/(S)-QE complexed system from the equilibrium trajectories. After T-test, the difference in the ∆Gbind value between these two QeH systems was statistically significant (nR= 351, nS = 201, t = 2.44, p = 0.015). We have modified it as “The total binding energy of the QeH-(R)-QE complexed system from the equilibrated conformations was calculated as -40.94 ± 3.7851 kcal·mol-1. However, for the QeH-(S)-QE complex, the ∆Gbind value of the equilibrium trajectory was calculated to be -38.95 ± 3.7882 kcal·mol-1. After T-test, the difference in the ∆Gbind value between these two QeH systems was statistically significant (nR= 351, nS = 201, t = 2.44, p = 0.015).” in the revised 2.2 section. Thank you for your comment again.

Comments 7: As a formatting decision, the materials and methods section comes after the results section. Is this the typical format for this journal? If not, it would be preferable in my opinion to have the materials and methods come before the results.

Response: Thank you for your comment. The section of Materials and methods is followed by results and discussion in accordance with format requirements of this journal.

Comments 8: Figure 1’s caption has root-moot-square fluctuation, which is presumably meant to be root-mean-square fluctuation.

Response: Thank you for pointing out this error. The “root-moot-square fluctuation” has been revised as “root-mean-square fluctuation”.

Comments 9: Additionally, the authors suggest figures C and D show the protein flexibility of QeH is lower in the QeH-R-QE system compared to QeH-S-QE, which is quite hard to see in the figure except for the tail.

Response: Thank you for your comment. Figures have been recreated, and the averaged RMSF value and standard deviation have also been calculated, clearly revealing the difference in flexibility between these two complexes. The revised results have been added as “By calculating the root-mean-square fluctuation (RMSF) of these two complexed systems, the flexibility change of QeH during the process of MD simulation was analyzed [21]. The averaged RMSF and standard deviation for all residues and after removing two terminal residues are listed in Table 3. As listed in Table 3 and shown in Figures 3C and D, the protein flexibility of QeH in the QeH-(R)-QE system was lower than that in the QeH-(S)-QE system.” in the revised 2.1 section.

Figure 3. Molecular dynamics (MD) simulations of the QeH-(R)-QE and QeH-(S)-QE complexed systems. The root-mean-square deviation (RMSD) and root-mean-square fluctuation (RMSF) curves of the QeH-(R)-QE (A and C) and QeH-(S)-QE (B and D) complexed systems as functions of simulation time during the MD runs.

System

Residue

Averaged RMSF

Standard Deviation

QeH-(R)-QE

all the residues

0.71

0.45

expect for the tail

0.69

0.34

QeH-(S)-QE

all the residues

0.87

1.16

expect for the tail

0.83

0.93

Table 3. The averaged RMSF and standard deviation for all the residues and except for the tail. (Measurement Unit: Å)

Comments 10: In Figure 2, the authors do not explain the labels in the two snapshots. These are presumably residues, but could this be clarified in the caption?

Response: Thank you for your suggestion. We have clarified it as “Figure 4. The ten representative snapshots of (R)-QE (A) and (S)-QE (B) superposed at their respective QeH active sites inside the interior of hydrophobic pocket during their MD runs. Key residues of QeH and two ligands are represented by stick models, and the residues (Tyr331, Tyr350, Trp351, Gly352 and Arg384 for the QeH-(R)-QE complex; Tyr189, Phe326, Glu328, Tyr350, Trp351 and Val354 for the QeH-(S)-QE complex) with their respective binding affinities over -1.0 kcal·mol-1 were marked by black labels.” in the revised manuscript.

Comment 11: For Table 2, the caption is split by the table, potentially because the table is too wide. Could the authors reformat this?

Response: Thank you for your suggestion. The width of Table 2 has been adjusted to be narrower in the revised manuscript.

Comment 12: In Figure 4, subfigure D is missing its label. Additionally, the legends for subfigures B and D are very hard to read, making it hard to discern the residue – conformational interactions.

Response: Thank you for your good suggestion. This figure has been redrawn in the revised manuscript, and the label of Figure has been added and the various interactions on the diagrams were marked larger.

Figure 6. The key interactions at the active sites of the representative conformations of the QeH-(R)-QE and QeH-(S)-QE complexes with equilibrium stabilization. The interactions derived from the representative conformation of the QeH-(R)-QE (A and B) and QeH-(S)-QE (C and D) complexes generated by the MD simulations were represented by dotted lines in different colors, and the unit of interaction distances was Å.

Comment 13: The English quality does need improving throughout the manuscript.   

Response: Thank you for your good suggestion. I'm sorry for writing these poor sentences. We checked throughout the manuscript and inappropriate sentences have been polished.

Round 2

Reviewer 1 Report

Comments and Suggestions for Authors

The authors have made considerable progress in revising the text based on my comments, which has notably improved the manuscript. Although the discussion has been expanded with the literature suggestions I provided, I observed that some of the key references I recommended in my previous review have still not been cited. Including these references would further enhance the completeness and rigor of the work, and I recommend addressing this to strengthen the manuscript further. After this modification, the article will be suitable for publication.

Reviewer 3 Report

Comments and Suggestions for Authors

The authors have answered all of my comments and suggestions thoroughly. I commend them for adding a substantial number of analyses to the original work. I believe the paper quality has been improved substantially as a result and I am happy to recommend it for publication in its current form.